# Robust Imitation Via Decision-Time Planning

## Abstract

The goal of imitation learning is to mimic expert behavior from demonstrations, without access to an explicit reward signal. A popular class of approach infers the (unknown) reward function via inverse reinforcement learning (IRL) followed by maximizing this reward function via reinforcement learning (RL). The policies learned via these approaches are however very brittle in practice and deteriorate quickly even with small test-time perturbations due to compounding errors. We propose *Imitation with Planning at Test-time* (IMPLANT), a new algorithm for imitation learning that utilizes decision-time planning to correct for compounding errors of any base imitation policy. In contrast to existing approaches, we retain both the imitation policy and the rewards model at decision-time, thereby benefiting from the learning signal of the two components. Empirically, we demonstrate that IMPLANT significantly outperforms benchmark imitation learning approaches on standard control environments and excels at zero-shot generalization when subject to challenging perturbations in test-time dynamics.

## 1 Introduction

The objective of imitation learning is to optimize agent policies directly from demonstrations of expert behavior. Such a learning paradigm sidesteps reward engineering, which is a key bottleneck for applying reinforcement learning (RL) in many real-world domains, *e.g.*, autonomous driving, robotics. In the presence of a finite dataset of expert demonstrations however, a key challenge with current approaches is that the learned policies can quickly deviate from intended expert behavior and lead to compounding errors at test-time (Osa et al., 2018). Moreover, it has been observed that imitation policies can be brittle and drastically deteriorate in performance with even small perturbations to the dynamics during execution (Christiano et al., 2016; de Haan et al., 2019).

A predominant class of approaches to imitation learning is based on inverse reinforcement learning (IRL) and involve successive application of two steps: (a) an IRL step where the agent infers the (unknown) reward function for the expert, followed by (b) an RL step where the agent maximizes the inferred reward function via a policy optimization algorithm. For example, many popular IRL approaches consider an adversarial learning framework (Goodfellow et al., 2014), where the reward function is inferred by a discriminator that distinguishes expert demonstrations from roll-outs of an imitation policy [IRL step] and the imitation agent maximizes the inferred reward function to best match the expert policy [RL step] (Ho & Ermon, 2016; Fu et al., 2017). In this sense, reward inference is only an intermediary step towards learning the expert policy and is discarded post-training of the imitation agent.

We introduce *Imitation with Planning at Test-time* (IMPLANT), a new algorithm for imitation learning that incorporates decision-time planning within an IRL algorithm. During training, we can use any standard IRL approach to estimate a reward function and a stochastic imitation policy, along with an additional value function. The value function can be learned explicitly or is often a byproduct of standard RL algorithms that involve policy evaluation, such as actor-critic methods (Konda & Tsitsiklis, 2000; Peters & Schaal, 2008). At decision-time, we use the learned imitation policy in conjunction with a closed-loop planner. For any given state, the imitation policy proposes a set of candidate actions and the planner estimates the returns for each of actions by performing fixed-horizon rollouts. The rollout returns are estimated using the learned reward and value functions. Finally, the agent picks the action with the highest estimated return and the process is repeated at each of the subsequent timesteps.

Conceptually, IMPLANT aims to counteract the imperfections due to policy optimization in the RL step by using the reward function (along with a value function) estimated in the IRL step for decision-time planning. We demonstrate strong empirical improvements using this approach over benchmark imitation learning algorithms in a variety settings derived from the MuJoCo-based benchmarks in OpenAI Gym (Todorov et al., 2012; Brockman et al., 2016). In default evaluation setup where train and test environments match, we observe that IMPLANT improves by 16.5% on average over the closest baseline.

We also consider transfer setups where the imitation agent is deployed in test dynamics that differ from train dynamics and the test dynamics are inaccessible to the agent during both training and decision-time planning. In particular, we consider the following three setups: (a) "causal confusion" where the agent observes nuisance variables in the state representation during training (de Haan et al., 2019), (b) motor noise which adds noise in the executed actions during testing (Christiano et al., 2016), and (c) transition noise which adds noise to the next state distribution during testing. In all these setups, we observe that IMPLANT consistently and robustly transfers to test environments with improvements of 35.2% on average over the closest baseline.

## 2 PRELIMINARIES

**Problem Setup.** We consider the framework of Markov Decision Processes (MDP) (Puterman, 1990). An MDP is denoted by a tuple $\mathcal{M} = (\mathcal{S}, \mathcal{A}, \mathcal{T}, p_0, r, \gamma)$, where $\mathcal{S}$ is the state space, $\mathcal{A}$ is the action space, $\mathcal{T} : \mathcal{S} \times \mathcal{A} \times \mathcal{S} \to \mathbb{R}_{\geq 0}$ are the stochastic transition dynamics, $p_0 : \mathcal{S} \to \mathbb{R}_{\geq 0}$ is the initial state distribution, $r : \mathcal{S} \times \mathcal{A} \to \mathbb{R}$ is the reward function, and $\gamma \in [0, 1)$ is the discount factor. We assume an infinite horizon setting. At any given state $s \in \mathcal{S}$, an agent makes decisions via a stochastic policy $\pi : \mathcal{S} \times \mathcal{A} \to \mathbb{R}_{\geq 0}$. We denote a trajectory to be a sequence of state-action pairs $\tau = (s_0, a_0, s_1, a_1, \cdots)$. Any policy $\pi$, along with MDP parameters, induces a distribution over trajectories, which can be expressed as $p_\pi(\tau) = p(s_0) \prod_{t=0}^{\infty} \pi(a_t|s_t) \mathcal{T}(s_{t+1}|s_t, a_t)$. The return of a trajectory is the discounted sum of rewards $R(\tau) = \sum_{t=0}^{\infty} \gamma^t r(s_t, a_t)$.

In reinforcement learning (RL), the goal is to learn a parameterized policy $\pi_\theta$ that maximizes the expected returns w.r.t. the trajectory distribution. Maximizing such an objective requires interaction with the underlying MDP for simulating trajectories and querying rewards. However, in many high-stakes scenarios, the reward function is not directly accessible and hard to manually design.

In imitation learning, we sidestep the availability of the reward function. Instead, we have access to a finite set of $D$ trajectories $\tau_E$ (a.k.a. demonstrations) that are sampled from an expert policy $\pi_E$. Every trajectory $\tau \in \tau_E$ consists of a finite length sequence of state and action pairs $\tau = (s_0, a_0, s_1, a_1, \cdots)$, where $s_0 \sim p_0(s)$, $a_t \sim \pi_E(\cdot|s_t)$, and $s_{t+1} \sim \mathcal{T}(\cdot|s_t, a_t)$. Our goal is to learn a parameterized policy $\pi_\theta$ which best approximates the expert policy given access to $\tau_E$. Next, we discuss the two major families of techniques for imitation learning.

### 2.1 BEHAVIORAL CLONING

Behavioral cloning (BC) casts imitation learning as a supervised learning problem over state-action pairs provided in the expert demonstrations (Pomerleau, 1991). In particular, we learn the policy parameters by solving a regression problem with states $s_t$ and actions $a_t$ as the features and target labels respectively. Formally, we minimize the following objective:

$$\ell_{BC}(\theta) := \sum_{(s_t, a_t) \in \tau_E} \|a_t - \pi_\theta(s_t)\|_2^2. \tag{1}$$

In practice, BC agents suffer from *distribution shift* in high dimensions, where small deviations in the learned policy quickly accumulate during deployment and lead to a significantly different trajectory distribution relative to the expert (Ross & Bagnell, 2010; Ross et al., 2011).

### 2.2 INVERSE REINFORCEMENT LEARNING

An alternative indirect approach to imitation learning is based on inverse reinforcement learning (IRL). Here, the goal is to infer a reward function for the expert and subsequently maximize the inferred reward to obtain a policy. For brevity, we focus on adversarial imitation learning approaches

to IRL (Goodfellow et al., 2014). These approaches represent the state-of-the-art in imitation learning and are also relevant baselines for our empirical evaluations.

Generative Adversarial Imitation Learning (GAIL) is an IRL algorithm that formulates imitation learning as an "occupancy measure matching" objective w.r.t. a suitable probabilistic divergence (Ho & Ermon, 2016). GAIL consists of two parameterized networks: (a) a policy network $\pi_\theta$ (generator) which is used to rollout agent trajectories (assuming access to transition dynamics), and (b) a discriminator $D_\phi$ which distinguishes between "real" expert demonstrations and "fake" agent trajectories. Given expert trajectories $\tau_E$ and agent trajectories $\tau_\theta$, the discriminator minimizes the cross-entropy loss:

$$\ell_{IRL}(\phi) := -\mathbb{E}_{\tau_E}\left[\log D_\phi(\tau_E)\right] - \mathbb{E}_{\tau_\theta}\left[\log(1 - D_\phi(\tau_\theta))\right]. \tag{2}$$

We then feed the discriminator output $-\log(1 - D_\phi(s, a))$ as the inferred reward function to the generator policy. The policy parameters $\theta$ can be updated via any regular policy optimization algorithm for the RL objective, *e.g.*, Ho & Ermon (2016) use the TRPO algorithm (Schulman et al., 2015). By simulating agent rollouts, GAIL seeks to match the full trajectory state-action distribution of the imitation agent with the expert as opposed to BC which greedily matches the conditional distribution of individual actions given the states. In practice, GAIL and its variants (Li et al., 2017; Fu et al., 2017) outperform BC but might need excessive interactions with the training environment for sampling rollouts during training. Crucially, both BC and IRL approaches tend to fail catastrophically in the presence of small perturbations and nuisances at test-time (de Haan et al., 2019).

## 3 THE IMPLANT FRAMEWORK

In the previous section, we showed that current IRL algorithms consider reward inference as an auxiliary task for imitation learning. Once the agents have been trained, the reward function is discarded and the learned policy is deployed.[1] Indeed, if the RL step post reward inference (*e.g.*, generator updates in GAIL) were optimal, then the reward function provides no additional information about the expert relative to the imitation policy. However, this is far from reality, as current RL algorithms can fail to return optimal solutions due to either representational or optimization issues. For example, there might be a mismatch in the architecture of the policy network and the expert policy, and/or difficulties in optimizing non-convex objective functions. In fact, the latter challenge gets exacerbated in adversarial learning scenarios due to a non-stationary reward.

Building off these observations, we propose *Imitation with Planning at Test-time* (IMPLANT), an imitation learning algorithm that employs the learned reward function for decision-time planning. The pseudocode for IMPLANT is shown in Algorithm 1. We can dissect IMPLANT into two sequential phases: a training phase and a planning phase.

**Training phase:** We can invoke any IRL algorithm, *e.g.*, GAIL to optimize for a stochastic imitation policy $\pi_\theta$ by optimizing for some inferred reward function $r_\phi$. Additionally, we also train a parameterized value function $V_\psi$ at this stage. Value function estimation is often a subroutine for many RL algorithms including those which are used to update the policy within the IRL setup, such as actor-critic methods (Konda & Tsitsiklis, 2000). For such algorithms, learning a value function does not incur any additional computation.

**Planning phase:** At decision-time, we use the imitation policy along with the learned value and reward functions for closed-loop planning. We build our planner based on model-predictive control (MPC) (Camacho & Alba, 2013). At any given state $s_t$ and time $t \geq 0$, we are interested in choosing action sequences for trajectories which maximizes the following objective:

$$a_t, a_{t+1}, \cdots, = \underset{a_t, a_{t+1}, \cdots}{\arg\max} R(\tau) = \sum_{t'=t}^{\infty} \gamma^{t'-t} r(s_{t'}, a_{t'}) \tag{3}$$

where $s_0 \sim p_0$ and $s_{t+1} \sim \mathcal{T}(\cdot | s_t, a_t)$ for all $t \geq 0$.

---

[1]In some cases, the reward function is transferred to a new environment and a new policy is learned using the reward function and *additional interactions* with the new environment. See Section 5 for further discussion.

---

**Algorithm 1:** *Imitation with Planning at Test-time* (IMPLANT)

---

1 **Input:** available dynamics $\hat{\mathcal{T}}$, expert demonstrations $\tau_E$, rollout budget $B$, rollout policy $\pi$,
   horizon $H$, test start state $s_0$

2 **Note:** For brevity, we omit relevant MDP parameters in the list of arguments

3 **Function** `Train(`$\tau_E$`)`:

4    |   Learn a policy $\pi_\theta$ and a reward function $r_\phi$ with any existing IRL algorithm given access to
         demonstrations $\tau_E$ , *e.g.*, GAIL

5    |   Estimate a value function $V_\psi$ for $\pi_\theta$

6    |   **return** $\pi_\theta, r_\phi, V_\psi$;

7 **Function** `Plan(`$s, \pi_\theta, V_\psi, r_\phi, \pi, H, B, \hat{\mathcal{T}}$`)`:

8    |   Set $s = s_0$

9    |   **while** *agent is alive* **do**

10    |   |   // Agent planning

11    |   |   Sample $B$ trajectories $\{\tau^{(1)}, \tau^{(2)}, ..., \tau^{(B)}\}$ of max length $H$ starting from $s$ using
          dynamics $\hat{\mathcal{T}}$; sample the first action $a_0^{(i)} \sim \pi_\theta$, and sample subsequent actions from $\pi$
          as $a_{>0}^{(i)} \sim \pi$, for $i \in \{1, 2, ..., B\}$

12    |   |   Estimate trajectory returns $\hat{R}_{\phi,\psi}(\tau^{(i)})$ using $V_\psi$ and $r_\phi$ (see Eq. 4)

13    |   |   Pick best action index $i^* = \arg\max_i \hat{R}_{\phi,\psi}(\tau^{(i)})$ and execute the best action $a_0^{(i^*)}$

14    |   |   // Environment feedback

15    |   |   Observe true reward $r(s, a_0^{(i^*)})$ and true next state $s \sim \mathcal{T}(\cdot | s, a_0^{(i^*)})$

16    |   **end**

---

This objective has also been applied for model-based RL with a learned dynamics model and black-box access to the rewards function (Nagabandi et al., 2018; Chua et al., 2018). Unlike the RL setting however, we do not know the reward function for imitation learning. The true dynamics model may be available for planning (*i.e.*, $\hat{\mathcal{T}} = \mathcal{T}$) as in Ho & Ermon (2016) or can be estimated from expert demonstrations or online interactions (Baram et al., 2016). Hence, we can do rollouts as before in regular model-based RL but need to rely on learned estimates for the reward function. In particular, we use the learned reward function $r_\phi$ up to a fixed horizon $H$ and a terminal value function $V_\psi$ thereafter to estimate the trajectory return as:

$$R(\tau) \approx \sum_{t'=t}^{t+H-1} \gamma^{t'-t} r_\phi(s_{t'}, a_{t'}) + \gamma^H V_\psi(s_H) := \hat{R}_{\phi,\psi}(\tau). \tag{4}$$

Substituting Eq. 4 in Eq. 3, we obtain a surrogate objective for optimization. To optimize this surrogate, we propose a variant of the random shooting optimizer (Richards, 2005) that works as follows. At the current state $s_t$, we first sample a set of $B$ candidate actions independently from the imitation policy. For each candidate action, we estimate a score based on their expected returns by performing rollout(s) of fixed-length $H$. The rollout policy $\pi$ from which we sample all subsequent actions could be random (potentially high variance) or the imitation policy $\pi_\theta$ (potentially high bias) or a mixture. In our experiments, we obtained consistently better performance with using $\pi_\theta$ as the rollout policy $\pi$. For each trajectory, we estimate its return via Eq. 4 and finally, pick the action with the largest return.

Consistent with the closed-loop nature of MPC, we repeat the above procedure at the next state $s_t$. Doing so helps correct for errors in estimation and optimization in the previous time step, albeit at the expense of additional computation. The algorithm has two critical parameters that induce similar computational trade-offs. First, we need to specify a budget $B$ for the total number of rollouts. The higher the budget, larger is our search space for the best action. Second, we need to specify a planning horizon $H$. For larger lengths, we need extra computation that also involves interactions with the dynamics of the environment and rely more on the learned reward function than the value function for estimating returns in Eq. 4. However, since the rollouts are independent, we can mitigate additional computational costs by parallelizing the rollouts. While this parallelization

Table 1: Average return of imitation learning algorithms on MuJoCo benchmarks.

|  | Hopper | HalfCheetah | Walker2d |
|---|---|---|---|
| Expert | 3570 | 891 | 3593 |
| BC | $127 \pm 85$ | $427 \pm 131$ | $258 \pm 262$ |
| BC-Dropout | $169 \pm 105$ | $542 \pm 275$ | $1622 \pm 861$ |
| GAIL | $3506 \pm 337$ | $954 \pm 282$ | $2780 \pm 1007$ |
| GAIL-Reward Only | $319 \pm 123$ | $5 \pm 117$ | $56 \pm 146$ |
| IMPLANT (ours) | $\mathbf{3633} \pm 50$ | $\mathbf{1193} \pm 143$ | $\mathbf{3360} \pm 442$ |

is indeed bottlenecked by the rollout with the largest horizon, in all of our experiments, we perform rollouts of fixed length and the horizon that corresponds to the optimal performance is relatively small ($10 \sim 50$). Thus, the gains due to parallelization are significant.

In the next section, we present our empirical validation that also investigates the effect of planning horizon on the performance of the algorithm in greater detail.

## 4 EXPERIMENTS

Our experiments aim to evaluate the performance of IMPLANT as a standalone imitation algorithm in two kinds of settings. First, we evaluate its performance in the default "no-transfer" setting, where the agent is trained and tested in the same environment. Second, we emphasize the robustness of IMPLANT by evaluating its zero-shot generalization performance in environments where the test dynamics are a perturbed version of the training dynamics. We consider 3 such perturbations: causal confusion (de Haan et al., 2019), motor noise (Christiano et al., 2016), and transition noise. We will describe each of these setups subsequently alongside the results. For all transfer settings, we only assume access to the training dynamics $\mathcal{T}_{train}$ and use it as $\hat{\mathcal{T}}$ for planning. At test-time, no additional interactions is allowed, nor do we have access to the test dynamics $\mathcal{T}_{test}$.

**Setup.** We evaluate our approach on MuJoCo enviroments in OpenAI Gym (Brockman et al., 2016): Hopper, HalfCheetah, and Walker2d. The expert data used for benchmarking imitation learning on these environments is publicly available[2]. We replicate the experimental setup of Ho & Ermon (2016) by fixing a limited number of expert trajectories used for training, as well as sub-sampling expert trajectories every 20 time steps. All results are averaged over 5 runs of each algorithm with different seeds. We provide further details in Appendix A.

**Baselines.** As we observed in Algorithm 1, IMPLANT can employ any IRL algorithm under the hood. For our experiments, we consider GAIL (Ho & Ermon, 2016) as the IRL algorithm of choice both as input for IMPLANT and consequently, as the closest baseline of interest. GAIL is amongst the current state-of-the-art methods for imitation learning; see Section 2.2 for a detailed description. For every environment, we report results for IMPLANT using a single set of hyperparameters for the rollout budget and planning horizon. We provide further details in Appendix A.

In addition, we also consider a Behavioral Cloning (BC) baseline; see Section 2.1 for a detailed description. Further, we also tested two variants of GAIL and BC that employ dropout (Srivastava et al., 2014) to demonstrate the limited utility of standard regularization techniques in countering the challenges due to low data and test noise. In fact, GAIL with dropout completely failed to learn in the adversarial setting on any of the environments; for brevity, we exclude it from presentation.

Last, we include a "GAIL-Reward Only" ablation baseline where we discard the imitation policy (generator) of GAIL during execution and instead, only use the inferred reward model (discriminator) in conjunction with a random policy for decision-time planning. This directly contrasts with the GAIL baseline, which by default only uses the generator. On the other hand, IMPLANT uses both the generator and discriminator for imitation via decision-time planning.

---

[2]https://github.com/openai/baselines

Table 2: Average return of imitation learning algorithms in causal confusion setting.

|  | Hopper | HalfCheetah | Walker2d |
|---|---|---|---|
| Expert | 3570 | 891 | 3593 |
| BC | $209 \pm 121$ | $331 \pm 141$ | $119 \pm 206$ |
| BC-Dropout | $162 \pm 108$ | $548 \pm 175$ | $700 \pm 433$ |
| GAIL | $579 \pm 484$ | $699 \pm 200$ | $613 \pm 465$ |
| GAIL-Reward Only | $515 \pm 302$ | $-82 \pm 79$ | $57 \pm 146$ |
| IMPLANT (ours) | $\mathbf{1717} \pm 1262$ | $\mathbf{827} \pm 375$ | $\mathbf{807} \pm 395$ |

## 4.1 IMITATION WITH LIMITED EXPERT TRAJECTORIES

With a relatively low number of expert trajectories, it has been shown by Ho & Ermon (2016) that GAIL can achieve near-expert performance in almost all these environments. We evaluate the performance of IMPLANT using the lowest number of expert trajectories tested in prior work. The results are shown in Table 1. We find that IMPLANT can achieve near-optimal performance on all these environments, including Walker2d where GAIL performs much worse than the expert. As expected, BC and BC and BC-Dropout perform poorly in this setting. GAIL-Reward Only exhibits the poorest performance suggesting the benefits of explicitly learning a parametric policy.

## 4.2 ZERO-SHOT TRANSFER SETTING: CAUSAL CONFUSION

de Haan et al. (2019) observed that imitation learning approaches are susceptible to *causal confusion*, *i.e.*, their performance deteriorates significantly in the presence of nuisance confounders in the state representation. To demonstrate this phenomena empirically, de Haan et al. (2019) further propose a challenging example setup in which the nuisance can be created by appending the agent's observation with its action from the previous time step. A standard imitation agent trained in this environment will learn to *copy* the previous action (since successive actions are highly correlated in expert demonstrations), falling prey to causal confusion. At test-time, the agent's performance drops drastically if the appended action is replaced by random noise (*i.e.*, the confounding is removed). We refer the reader to de Haan et al. (2019) for further details and analysis.

We now benchmark the zero-shot test performance of IMPLANT under the same setup in Table 2. While all baselines, including GAIL, fail drastically due to the confounding nuisance, IMPLANT is significantly more robust in all environments. We can visualize the agent performance qualitatively in Figure 3. Note that we provided the IMPLANT agent access to only the confounded dynamics for decision-time planning. The algorithm is hence zero-shot, unlike the proposed solutions of Fu et al. (2017) and de Haan et al. (2019) which require further interactions with the non-confounded test environment for recovery.

## 4.3 ZERO-SHOT TRANSFER SETTING: OBSERVATION AND ACTION NOISE

Next, we consider two kinds of noisy perturbations motivated by real-world applications in sim2real.

First, we perturb the intended actions via *motor noise* (Christiano et al., 2016), *e.g.*, due to imperfect hardware, a real robot might execute a noisy version of the action proposed by the agent. We implement this scenario by adding independent Gaussian noise to each dimension of the executed action at test-time, *i.e.*, $\epsilon_{\text{action}} \overset{i.i.d.}{\sim} \mathcal{N}(0, \sigma^2)$ and we vary the noise stddev $\sigma \in [0.1, 0.2, 0.5, 1.0]$.

Second, we consider *transition noise* due to an imperfect dynamics model for a simulator that may not be able to account for perturbations due to drag or friction. Hence, we specify the test-time dynamics to be a perturbed noisy version of the training dynamics. Similar to motor noise, we sample the transition noise as $\epsilon_{\text{transition}} \overset{i.i.d.}{\sim} \mathcal{N}(0, \sigma^2)$ with $\sigma \in [0.001, 0.002, 0.005, 0.01]$.

For ease of visualization, we show the normalized performance of the different algorithms in Figure 2. See Appendix B for raw absolute results. We also include another competitive baseline "GAIL-Expert-Noise" relevant to this scenario that artificially adds independent noise to the demonstration data for every gradient update during GAIL training. For a very high noise level, any

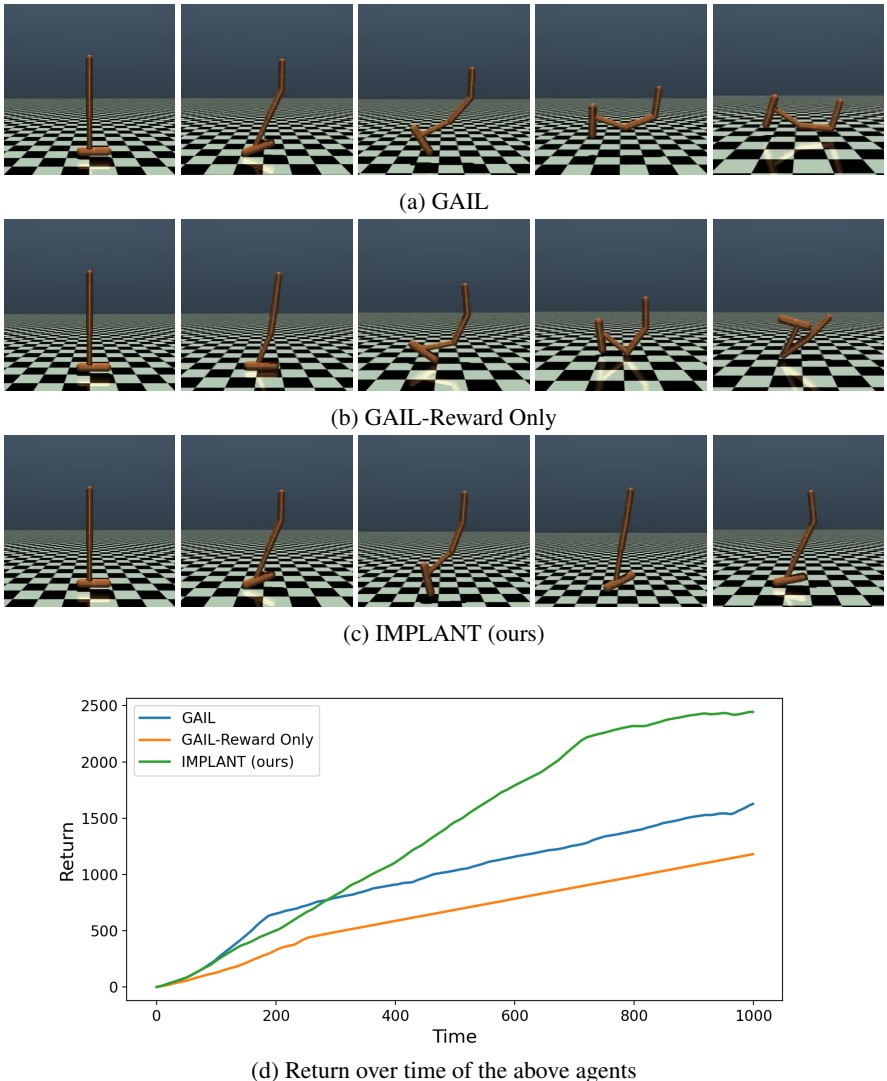

(a) GAIL

(b) GAIL-Reward Only

(c) IMPLANT (ours)

(d) Return over time of the above agents

Figure 1: Trajectory visualization for Hopper environment in causal confusion setup at test-time. While all agents start from the same state, only IMPLANT can effectively hop forward. All agents are trained in the confounded setting and tested in the non-confounded setting.

algorithm will naturally deteriorate in performance due to significant shift in training and testing environments. More importantly, for modest noise levels, we find that IMPLANT outperforms the baselines in almost all cases, highlighting its robustness.

## 4.4 EFFECT OF PLANNING HORIZON

Finally, we analyze the effect of planning horizon on IMPLANT performance in the same setup as Section 4.1. Specifically, we vary the planning horizon $H \in [0, 10, 50, 100]$ for a rollout budget $B = 10$. The normalized performance curves are shown in Figure 3. When the planning horizon is 0, we only rely on the terminal value function for estimating returns. Conversely, for large planning horizons (*e.g.*, $H = 100$), the returns are dominated by rewards accumulated at every time step. We observe that picking neither a very large horizon ($h \geq 100$) nor a very small one ($h = 0$) results in optimal performance, suggesting imperfections in both the

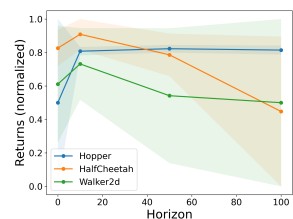

Figure 3: Effect of varying planning horizon $H$ on IM-PLANT performance.

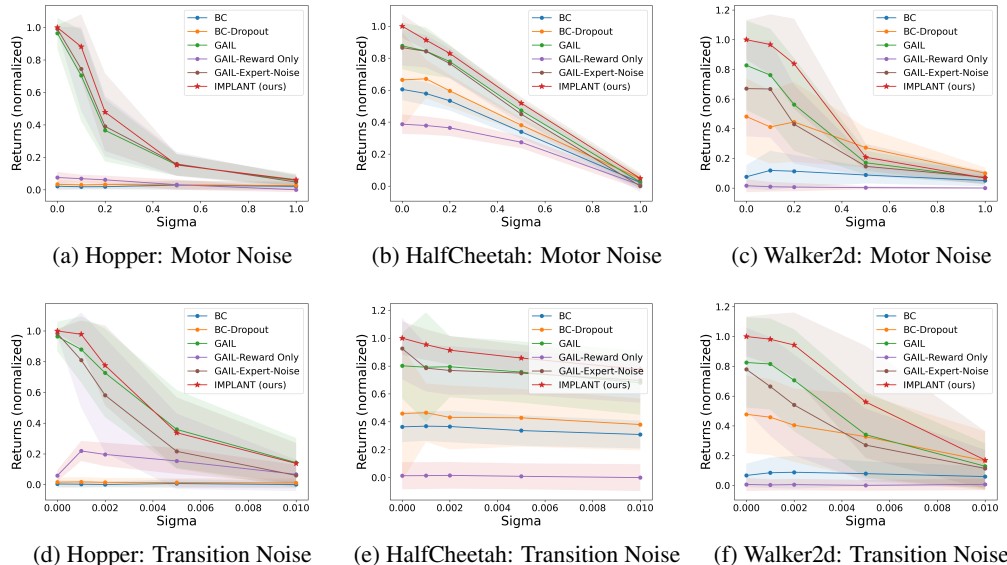

(a) Hopper: Motor Noise
(b) HalfCheetah: Motor Noise
(c) Walker2d: Motor Noise

(d) Hopper: Transition Noise
(e) HalfCheetah: Transition Noise
(f) Walker2d: Transition Noise

Figure 2: Average return of imitation learning algorithms on motor and transition noise settings.

learned reward and value functions and the sweet-spot for the planning horizon is typically between the extremes.

## 5 DISCUSSION & RELATED WORK

Traditionally, algorithms for imitation learning fall into one of two categories. They are either completely model-free during both training and execution, as in behavioral cloning and its variants (Pomerleau, 1991; Ross et al., 2011). Alternatively, they are model-based in the sense that they utilize dynamics and (inferred) rewards models during training, but are model-free during execution, as in inverse reinforcement learning (Ng et al., 2000; Ratliff et al., 2006; Ziebart et al., 2008). Our work introduces a novel model-based perspective to imitation learning where the reward and transition models are used *both* during training and execution. Borrowing the terminology from Sutton & Barto (2018), the use of models for the MDP during training and execution are also referred to as *background* and *decision-time* planning respectively.

While imitation via background planning has showed immense promise for control in complex environments (Abbeel & Ng, 2004; Ratliff et al., 2009; Ho & Ermon, 2016; Choudhury et al., 2018), we showed that decision-time planning in IMPLANT can further improve the data efficiency and robustness of the learned policies. There have also been several alternate attempts for characterizing and enhancing the robustness of imitation policies. For example, Fu et al. (2017) seek robustness in the sense of recovering the true reward function via adversarial imitation learning and transfer the inferred reward function to external dynamics in the non-zero shot setting. A significant body of work also considers IRL approaches that can accurately capture the uncertainty in the reward function for safe deployment (Zheng et al., 2014; Brown et al., 2018; Huang et al., 2018; Lacotte et al., 2019; Brown et al., 2020). While these utility-based notions are distinct from ours, they are complementary approaches to robustness that could be combined with IMPLANT in future work.

Given the synergies between generative modeling and imitation learning as exemplified in GAIL (Ho & Ermon, 2016), improvements in the former often translate into improved imitation, *e.g.*, the use of autoencoder embeddings to improve diversity (Wang et al., 2017), better loss functions and architectures for stable GAN/GAIL training (Pfau & Vinyals, 2016; Kuefler et al., 2017; Li et al., 2017), etc. These modifications are conceptually complementary to the key contribution of IMPLANT to incorporate decision-time planning and are likely to further boost our performance. In fact, decision-time planning in IMPLANT can be viewed as filtering of trajectories sampled from the policy network. This is similar to recent work in using importance weighting for improving sample quality of a

generative model (Grover & Ermon, 2017; Azadi et al., 2018; Grover et al., 2019). However, our solution is tailored towards sequential decision making and deterministically picks the best outcome in line with model predictive control, unlike importance weighting filters.

## 6 CONCLUSION

We presented *Imitation with Planning at Test-time* (IMPLANT), a new algorithm for imitation learning that uses decision-time planning to mitigate compounding errors of any base IRL algorithm. Unlike existing approaches, IMPLANT is truly model-based in the sense of utilizing the inferred rewards and dynamics model both during training and execution. We demonstrated that IMPLANT matches or outperforms existing benchmark imitation learning algorithms with very few expert trajectories. Finally, we empirically demonstrated the robustness of IMPLANT via its impressive performance at zero-shot generalization in several challenging perturbation settings.

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

Table 3: Detailed information on environments

| Environment | Observation space | Action space |
|---|---|---|
| Hopper | Box(11, ) | Box(3, ) |
| HalfCheetah | Box(17, ) | Box(6, ) |
| Walker2d | Box(17, ) | Box(6, ) |

Table 4: Detailed information on expert data used in training

| Environment | # of trajectories | # of state-action pairs |
|---|---|---|
| Hopper | 4 | 200 |
| HalfCheetah | 4 | 200 |
| Walker2d | 20 | 1000 |

## A    ADDITIONAL EXPERIMENTAL DETAILS AND SETUPS

### A.1    ENVIRONMENTS AND EXPERT DATA

As mentioned above, we consider 3 continuous tasks from OpenAI Gym (Brockman et al., 2016) simulated with MuJoCo (Todorov et al., 2012): Hopper, HalfCheetah, and Walker2d. We acquire expert data from OpenAI Baselines (Dhariwal et al., 2017). Table 3 lists more detailed information about each environment, and Table 4 contains information about the expert demonstrations we use for training all of the agents.

### A.2    HYPERPARAMETERS AND NETWORK ARCHITECTURES

We use a 2-layer MLP with *tanh* activations and 64 hidden units for all of our policy networks. For BC, we use a learning rate of $10^{-4}$ across all environments. We use a dropout rate of 0.2 for our BC-Dropout agent. The hyperparameters of GAIL are listed in Table 5. For our IMPLANT agent, we directly utilize the value function, reward function, and policy from a trained GAIL agent. In all settings, we choose $B = 20$ and $H = 50$ for Hopper, $B = 2$ and $H = 10$ for HalfCheetah and Walker2d to plan.

## B    ADDITIONAL RESULTS

The complete results for Section 4.3 can be found in Table 6, 7, 8.

Table 5: Detailed information on GAIL's hyperparameters

| Parameters | Hopper | HalfCheetah | Walker2d |
|---|---|---|---|
| Discriminator network | 100-100 MLP | 100-100 MLP | 100-100 MLP |
| Discriminator entropy coeff. | 0.01 | 0.01 | 0.01 |
| Batch size | 1024 | 1024 | 1024 |
| Max kl | 0.01 | 0.01 | 0.01 |
| CG steps/damping | 10, 0.01 | 10, 0.1 | 10, 0.1 |
| Entropy coeff. | 0.0 | 0.0 | 0.0 |
| Value fn. steps/step size | 3, 3e-4 | 5,1e-3 | 5,1e-3 |
| Generator steps | 3 | 3 | 3 |
| Discriminator steps | 1 | 1 | 1 |
| $\lambda$ | 0.98 | 0.97 | 0.97 |
| $\gamma$ | 0.99 | 0.995 | 0.995 |

Table 6: Raw results of Figure 2 in Hopper environment

(a) Hopper with motor noise

| Sigma | 0.0 | 0.1 | 0.2 | 0.5 | 1.0 |
|---|---|---|---|---|---|
| BC | $127 \pm 85$ | $114 \pm 64$ | $117 \pm 61$ | $179 \pm 105$ | $123 \pm 82$ |
| BC-Dropout | $169 \pm 105$ | $156 \pm 85$ | $163 \pm 95$ | $158 \pm 89$ | $142 \pm 94$ |
| GAIL | $3506 \pm 337$ | $2572 \pm 1008$ | $1360 \pm 687$ | $598 \pm 243$ | $252 \pm 128$ |
| GAIL-Reward Only | $319 \pm 123$ | $293 \pm 87$ | $268 \pm 92$ | $160 \pm 113$ | $49 \pm 54$ |
| GAIL-Expert-Noise | $3602 \pm 46$ | $2716 \pm 921$ | $1449 \pm 667$ | $\mathbf{618} \pm 264$ | $218 \pm 136$ |
| IMPLANT (ours) | $\mathbf{3633} \pm 50$ | $\mathbf{3209} \pm 714$ | $\mathbf{1764} \pm 856$ | $596 \pm 234$ | $\mathbf{269} \pm 130$ |

(b) Hopper with transition noise

| Sigma | 0.0 | 0.001 | 0.002 | 0.005 | 0.01 |
|---|---|---|---|---|---|
| BC | $127 \pm 85$ | $123 \pm 81$ | $116 \pm 65$ | $137 \pm 98$ | $114 \pm 84$ |
| BC-Dropout | $169 \pm 105$ | $175 \pm 113$ | $165 \pm 99$ | $161 \pm 98$ | $151 \pm 101$ |
| GAIL | $3506 \pm 337$ | $3209 \pm 756$ | $2672 \pm 1012$ | $\mathbf{1377} \pm 901$ | $\mathbf{616} \pm 563$ |
| GAIL-Reward Only | $319 \pm 123$ | $884 \pm 230$ | $804 \pm 269$ | $665 \pm 296$ | $349 \pm 215$ |
| GAIL-Expert-Noise | $3576 \pm 36$ | $2966 \pm 1089$ | $2160 \pm 1311$ | $875 \pm 865$ | $323 \pm 357$ |
| IMPLANT (ours) | $\mathbf{3633} \pm 50$ | $\mathbf{3557} \pm 313$ | $\mathbf{2844} \pm 915$ | $1301 \pm 810$ | $598 \pm 467$ |

Table 7: Raw results of Figure 2 in HalfCheetah environment

(a) HalfCheetah with motor noise

| Sigma | 0.0 | 0.1 | 0.2 | 0.5 | 1.0 |
|---|---|---|---|---|---|
| BC | $427 \pm 131$ | $377 \pm 139$ | $289 \pm 118$ | $-87 \pm 81$ | $-703 \pm 66$ |
| BC-Dropout | $542 \pm 275$ | $555 \pm 232$ | $409 \pm 194$ | $-6 \pm 85$ | $-655 \pm 61$ |
| GAIL | $954 \pm 282$ | $892 \pm 287$ | $764 \pm 200$ | $172 \pm 121$ | $-697 \pm 82$ |
| GAIL-Reward Only | $5 \pm 117$ | $-11 \pm 110$ | $-39 \pm 100$ | $-214 \pm 72$ | $-732 \pm 53$ |
| GAIL-Expert-Noise | $933 \pm 216$ | $888 \pm 195$ | $740 \pm 179$ | $127 \pm 124$ | $-745 \pm 74$ |
| IMPLANT (ours) | $\mathbf{1193} \pm 143$ | $\mathbf{1025} \pm 115$ | $\mathbf{863} \pm 94$ | $\mathbf{260} \pm 72$ | $\mathbf{-652} \pm 65$ |

(b) HalfCheetah with transition noise

| Sigma | 0.0 | 0.001 | 0.002 | 0.005 | 0.01 |
|---|---|---|---|---|---|
| BC | $427 \pm 131$ | $433 \pm 126$ | $430 \pm 136$ | $395 \pm 131$ | $361 \pm 120$ |
| BC-Dropout | $542 \pm 275$ | $549 \pm 252$ | $509 \pm 276$ | $505 \pm 249$ | $447 \pm 223$ |
| GAIL | $954 \pm 282$ | $942 \pm 472$ | $946 \pm 263$ | $900 \pm 264$ | $808 \pm 276$ |
| GAIL-Reward Only | $5 \pm 117$ | $6 \pm 116$ | $8 \pm 115$ | $0 \pm 112.$ | $-11 \pm 116$ |
| GAIL-Expert-Noise | $1103 \pm 265$ | $934 \pm 201$ | $914 \pm 209$ | $891 \pm 186$ | $828 \pm 192$ |
| IMPLANT (ours) | $\mathbf{1193} \pm 143$ | $\mathbf{1137} \pm 132$ | $\mathbf{1089} \pm 123$ | $\mathbf{1021} \pm 115$ | $\mathbf{923} \pm 103$ |

Table 8: Raw results of Figure 2 in Walker2d environment

(a) Walker2d with motor noise

| Sigma | 0.0 | 0.1 | 0.2 | 0.5 | 1.0 |
|---|---|---|---|---|---|
| BC | $258 \pm 262$ | $402 \pm 444$ | $384 \pm 368$ | $298 \pm 191$ | $172 \pm 128$ |
| BC-Dropout | $1622 \pm 861$ | $1386 \pm 826$ | $1498 \pm 901$ | $\mathbf{918} \pm 450$ | $\mathbf{339} \pm 129$ |
| GAIL | $2780 \pm 1007$ | $2560 \pm 1047$ | $1893 \pm 1045$ | $576 \pm 272$ | $234 \pm 133$ |
| GAIL-Reward Only | $56 \pm 146$ | $33 \pm 109$ | $25 \pm 94$ | $13 \pm 36$ | $4 \pm 13$ |
| GAIL-Expert-Noise | $2253 \pm 1081$ | $2245 \pm 1098$ | $1448 \pm 891$ | $495 \pm 221$ | $247 \pm 144$ |
| IMPLANT (ours) | $\mathbf{3360} \pm 442$ | $\mathbf{3251} \pm 682$ | $\mathbf{2816} \pm 1019$ | $701 \pm 301$ | $228 \pm 148$ |

(b) Walker2d with transition noise

| Sigma | 0.0 | 0.001 | 0.002 | 0.005 | 0.01 |
|---|---|---|---|---|---|
| BC | $258 \pm 262$ | $318 \pm 338$ | $329 \pm 365$ | $298 \pm 274$ | $233 \pm 230$ |
| BC-Dropout | $1622 \pm 861$ | $1559 \pm 848$ | $1380 \pm 819$ | $1123 \pm 937$ | $584 \pm 658$ |
| GAIL | $2780 \pm 1007$ | $2746 \pm 1015$ | $2383 \pm 1136$ | $1168 \pm 958$ | $464 \pm 511$ |
| GAIL-Reward Only | $56 \pm 146$ | $46 \pm 124$ | $53 \pm 128$ | $36 \pm 120$ | $59 \pm 129$ |
| GAIL-Expert-Noise | $2627 \pm 927$ | $2244 \pm 1098$ | $1832 \pm 1125$ | $935 \pm 812$ | $414 \pm 412$ |
| IMPLANT (ours) | $\mathbf{3360} \pm 442$ | $\mathbf{3299} \pm 558$ | $\mathbf{3170} \pm 725$ | $\mathbf{1899} \pm 1249$ | $\mathbf{600} \pm 644$ |

