# OpenReview forum: "Robust Imitation via Decision-Time Planning"
_ICLR.cc/2021/Conference — Reject_

### Official Review · AnonReviewer1 · 2020-10-21
**Good presentation, but its hard to name the contribution**

**Rating:** 3
**Confidence:** 5

**Review:**

__Summary__
The paper proposes to combine imitation learning (GAIL) with model predictive control to improve on the policy, especially when the system dynamics are noisier at test time.
Here, MPC uses the reward function and value function learned by GAIL and assumes access to a model of the test-dynamics.
The proposed approach, IMPLANT, is compared to GAIL and behavioral cloning on MuJoCo experiments in a standard imitation learning setting and for three slightly modified test scenarios that add either action noise, transition noise, or causal confusing which adds the last actions to the state space during training and replaces this information by noise at test time.


__Strong points / weak points__
+ (+) The paper is well-written and very easy to follow
+ (+) Using MPC on a reward function that was learned by IRL can be reasonable
- (-) Lack of contribution / novelty
- (-) Using the GAIL reward doesn't seem sound. An actual IRL method should be used
- (-) additional assumptions compared to competitors (access to noisy model, computational tractability of online planning) reduce practicability and are not regarded during the comparisons


__Recommendation__
I recommend rejecting the submission because I can't find a noteworthy contribution.


__Supporting Arguments__
- *Lack of contribution:* I don't want to be gatekeeping, but I really don't see any noteworthy contribution. The standard approach for apprenticeship learning is to learn a reward function by IRL and then optimize this reward function to learn a policy. There is no requirement to use the same algorithm to optimize the reward that used for inferring it. It is not surprising that reoptimizing the reward performs better in face of dynamic perturbations compared to a direct policy transfer; after all, this is one of the main motivations for learning a reward function. It is also not surprising that a model-based MPC approach based on the policy can perform better than directly using the parametric policy, even when there are no changes in the dynamics. So, I actually don't even see the research question here. There can be a small contribution due to the empirical evaluation, however, the chosen baselines perform direct policy transfer and, thus, do not seem that interesting.

- *Using GAIL for IRL:* The paper repeatably refers to GAIL as an IRL approach. However, GAIL does not infer a reward function in accordance with the problem formulation of IRL. The "reward function" used by GAIL is only valid for small policy updates of the current policy (generator). *Maximizing* this reward function will in general not result in any reasonable policy. For optimal policy and discriminator, the reward function would even be constant. So using this reward function for MPC doesn't seem sound. The evaluation did show that optimizing the reward function via MPC can be beneficial when using a small enough horizon and when sampling from the parametric policy, but this is not that surprising since the MPC control will in that case tend to remain close to the parametric policy.

- *Additional assumptions:* The paper argues that IMPLANT is "zero-shot, unlike the proposed solutions of Fu et al. (2017) and de Haan et al. (2019) which require further interactions" with the test environment. However, I would argue that MPC should be treated like a reinforcement learning method here; it also requires interactions with the test environment (i.e., sampling from the dynamics model). I don't see why it would not be fair to use a policy-parametric RL algorithm (e.g. AIRL + reoptimizing) instead of MPC for optimizing the reward. Actually, I think that the test setting would then still favor MPC since it requires costly online optimization which is not always feasible in practice.


__Questions__
1. Are there problem settings where IMPLANT is applicable, but IRL+RL (e.g. AIRL + reoptimizing) is not?
2. The paper mentions that a random rollout policy for MPC performs worse than using the learned policy. Is this also the case for a very large number of rollouts? Is the resulting control also worse in terms of the learned reward function or only w.r.t. the true reward function?


__Additional Feedback__
My main concern is that the current submission has too little contribution and, thus, I think that more research needs to be done before publishing a paper on this topic. Performing MPC after IRL seems way too little as a paper story. It's hard to suggest a direction for further research because I do not see the research question to start with. Regarding the combination of IRL and MPC, it might be interesting to investigate using MPC within the IRL loop. For example, it can be challenging to learn reward functions for multimodal demonstration based on a parametric (and usually unimodal) policy.

The current paper could be improved by using an IRL reward and by comparing the approach with IRL+reoptimizing. However, at least for me, this would not affect my recommendation because these changes do not address my main concern.

I hope this review is not too discouraging; the paper does have strong points in the presentation.

---

> ### Author Response · Authors · 2020-11-19
> **Response to reviewer questions and feedback**
>
> We thank the reviewer for the extensive comments! We would like to clarify some key points with regard to our problem setting. Most importantly, we do not require knowledge of the test-time dynamics for planning. We have further emphasized this point in the revised draft. In light of this clarification, we address specific concerns below.
> * Lack of contribution:
>   * RE: “no requirement to use the same algorithm to optimize learned reward:"
> Yes, we agree. IMPLANT is not restricted to such algorithms either.
>   * RE: “not surprising that reoptimizing the reward performs better performs better in perturbations"
> Yes, we agree. However, this simple reoptimization scheme requires knowledge of the test-time perturbations. This is *not* the case for IMPLANT; we are not assuming knowledge of the perturbed test-time dynamics for reoptimization.
>   * RE: “not surprising that model-based MPC based on the policy perform better"
> We respectfully disagree. It is not guaranteed that model-based approaches (or hybrids of model-free and model-based approaches) outperform purely model-free approaches in terms of asymptotic performance. Quite the contrary, model-based RL approaches are generally worse, and improving these approaches is an active area of research [1]. For instance, the current SOTA is SAC which is model-free and generally outperforms the SOTA model-based algorithm (MBPO).
> * Using GAIL for IRL
>   * RE: “for optimal policy and discriminator, the reward would be constant, so this reward for MPC doesn’t seem sound”
> We agree with the theoretical argument, but this is not true in practice. Empirically, neither the generator nor the discriminator is perfect, especially when trained with limited expert trajectories (Sec 4.1) or in presence of nuisance variables and noise (sec 4.2). With regards to soundness, note that IMPLANT with an optimal policy and discriminator (and hence, constant discriminator output for reward) reduces to GAIL. Since GAIL is assumed to be optimal here, so is IMPLANT by corollary.
> More importantly, IMPLANT can be employed over any existing IRL approach. To demonstrate this empirically, we tested IMPLANT on a recent state-of-the-art non-adversarial IRL algorithm,  PWIL [2]. The results below show that our algorithm still outperforms all of the baselines in Hopper both in the non-transfer setting (sigma=0.0) and in the presence of motor noise perturbation at test-time:
> |sigma|0.0|0.1|0.2 |0.5|1.0|
> |----------------|------------|-------------|------------|-----------|----------|
> |PWIL |3552 ± 200|2189 ± 1236| 935 ± 706|**393 ± 119**| 218 ± 86|
> |IMPLANT (ours) |**3563 ± 87**|**2637 ± 1140**|**1110 ± 817**|390 ± 103|**232 ± 82**|
> * Additional assumptions, unfair comparison with AIRL.
> We apologize for the confusion, but we’ve modified the paper to clarify that we do not require interactions with the *test* environment. Instead, we use the training dynamics to plan at test-time, and even with a mismatched dynamics model, our approach still achieves higher performance than a direct policy transfer approach. Thus, our approach is zero-shot. In contrast,  AIRL reoptimizes the learned reward function at test-time to perform RL, which indeed requires interactions with the test environment.
> * Questions:
>   * Q: Settings where IMPLANT is applicable, but IRL+RL (e.g. AIRL + reoptimizing) is not?
> A: As clarified above, we assume knowledge of the training dynamics, but not test dynamics, whereas IRL+RL reoptimizing requires interactions in test dynamics. Indeed this is what makes our transfer setting zero-shot. Our problem setting largely resembles sim2real transfer, in which the dynamics of a simulator could be easy to specify, but real-world interactions are costly.
>   * Q: "(Using a random policy performs worse)... Is this also the case for a very large number of rollouts? Is the resulting control also worse in terms of the learned reward function or only w.r.t. the true reward function?"
> A: We conducted more experiments using random rollout policy for MPC with 100, 500, 1000 rollouts (as compared to 2, 10, 20 using learned policies). We found that increasing rollouts results in better performance w.r.t learned reward but not w.r.t the true reward, with details shown below:
> Hopper-v2, fixing horizon h to be 10 and varying budget b
> | Budget|2|10|20|100|500|1000|
> |:---------------------------------|:---------:|:---------:|:---------:|:---------:|:---------:|:---------:|
> |Performance w.r.t. true reward|206 ± 120|273 ± 123|**281 ± 121**|275 ± 125|270 ± 113|278 ± 116|
> |Performance w.r.t. learned reward|886 ± 137|884 ± 68|892 ± 135|906 ± 57|964 ± 125|**997 ± 133**|
>
> We hope that the above comments can clarify our setting and show the contributions of our work. We appreciate the reviewer’s feedback and welcome any further comments.
>
> REFERENCES
> * [1] Tingwu  Wang et al., Benchmarking model-based reinforcement learning, 2019.
> * [2] Robert Dadashi et al., Primal wasserstein imitation learning, 2020.

---

> > ### Comment · AnonReviewer1 · 2020-11-20
> > **Thanks for clarifications**
> >
> > * Model-Based vs. Model-Free: I didn't say that model-based RL performs better than model-free. I said that it is not surprising that  model based _MPC_ (based on the same model that was used for learning the policy) and leveraging the learned policy performs better than directly applying the learned policy.
> >
> > * Thank you for clarifying that the training-dynamics are used for MPC, I indeed assumed that the test-dynamics are provided to MPC. I'm now much more positive about the fairness of the evaluations.
> >
> > * However, I'm still concerned about the novelty (using MPC with a learned reward function is a very marginal contribution) and the soundness (Using the GAIL reward makes little sense in my opinion and the additional experiments with random MPC-policy seem to confirm this; Why not use an actual IRL method for learning a reward function?)

---

> > > ### Author Response · Authors · 2020-11-23
> > > **Further clarifications to reviewers questions**
> > >
> > > Thank you for the comments! We greatly appreciate your active involvement in providing us additional feedback during the discussion phase.
> > >
> > > * Re: Model-Based MPC
> > > Thanks for the clarification about your original question. We however still believe that model-based MPC is not trivially guaranteed to outperform the learned policy. This is because the model (for IRL, this is the inferred rewards function) can be imperfect and hence, planning using the imperfect model is not guaranteed by any means to improve performance. Moreover, we are not aware of any existing work in imitation learning that employs this approach to improve asymptotic performance.
> > >
> > > * Re: novelty
> > > There are a few points we would like to mention here:
> > >   * As mentioned above, we believe that the improvements shown in our paper are not obvious---if it were the case, they would have likely been included as part of existing benchmarking. Consequently, we are making a deeper point about the role of reward inference for imitation learning i.e., reward inference is not just a subroutine for policy learning; by including it at test-time during planning, it implicitly becomes a part of the learned policy.
> > >   * We view the simplicity of our approach as a desirable feature. With less moving parts, IMPLANT is easier to reproduce and replicate as a key component of any base IRL algorithm.
> > >   * We especially believe that the empirical utility of the zero-shot results are non-trivial. For example, causal confusion (de Haan et al, 2019, oral presentation at NeurIPS 2019) is a big open challenge for imitation learning and significantly degrades the robustness of imitation learning algorithms; we have demonstrated significant improvements of 77% on average over the closest baseline for this problem via decision-time planning *without any targeted interventions*.
> > >
> > >
> > > * Re: soundness of GAIL / other IRL algorithms
> > > We would like to make a couple of points here:
> > >   * IRL is fundamentally an under-constrained problem, for which many reward functions can explain the same demonstration behavior [1]. Any IRL algorithm hence infers a proxy reward function; these proxies usually do not guarantee identifiability of the reward function (except under unrealistic assumptions) but yet are effective in practice [2]. GAIL is no exception; however we used it in our experiments because it is the most popular adversarial IL algorithm in use today, and empirically showed that its proxy reward function is suitable for MPC planning.
> > >   * In our first round of responses above, we also performed some additional experiments and trained PWIL---which is a SOTA non-adversarial IRL approach that does not have the issue of GAIL and its variants---and observed similar improvements. We hope these experiments provide evidence suggesting the general-purpose utility of our approach. If there are any specific recommendations of other IRL algorithms, please let us know and we will be happy to include a more extensive evaluation with those algorithms in the final version!
> > >
> > > [1] Andrew  Y  Ng,  Stuart  J  Russell,  et al. Algorithms for inverse reinforcement learning.
> > > [2] Saurabh Arora, Prashant Doshi. A Survey of Inverse Reinforcement Learning: Challenges, Methods and Progress, 2020.
> > >
> > > We hope that the reviewer will consider upgrading the original score in light of our earlier and current set of responses. Please let us know if you have any other questions.

---

### Official Review · AnonReviewer2 · 2020-10-28
**Interesting idea for tacking test-time perturbation in imitation learning, with limited testing environments.**

**Rating:** 6
**Confidence:** 3

**Review:**

#######################################################################

Summary:

The paper considers imitation learning problem in the presence of small perturbations and nuisances at test-time. In particular, it proposes Imitation with Planning at Test-time (IMPLANT), a new algorithm for imitation learning that incorporates decision-time planning within an inverse reinforcement learning algorithm. To counteract the imperfection due to policy optimization in RL step, IMPLANT uses the reward function estimated in the IRL step for decision-time planning. The effectiveness of the proposed method has been empirically evaluated on two kinds of setups, i.e., the default ‘no-transfer’ setting and the ‘transfer’ setting where the test dynamics is a perturbated version of the training dynamics.

#######################################################################

Reasons for score:

Overall, I vote for a weak acceptance. I like the idea of using test-time planning for tackling the test-time perturbation in imitation learning. My major concern is about the clarity of the paper and some additional environments (see cons below). Hopefully the authors can address my concern in the rebuttal period.

#######################################################################

Pros:

1. The paper takes one practical issue of imitation learning: test-time perturbation.

2. For me, the proposed IMPLANT method is novel for the issue of test-time perturbation in imitation learning. Specifically, IMPLANT builds on top of GAIL, and learns an additional value estimation during training, and utilizes the estimated reward function, value function for closed-loop planning based on model-predictive control (MPC). The design is interesting.

3. This paper provides comprehensive experiments, including both qualitative analysis and quantitative results, to show the effectiveness of the proposed framework. In particular, IMPLANT outperforms the BC and GAIL-based baselines in the ‘non-transfer’ setting of imitation with limited expert trajectories, and in the transfer setting with causal confusion, motor noise and transition noise.

#######################################################################

Cons:

1. For the motivation, it would be better to provide more details about it, which seems not very clear to me. Particularly, it is unclear why the planning is introduced, and why such design can address the issue studied in the paper. Additionally, since this paper claims any IRL methods can be used in the training time, I suggest the authors to showcase and discuss the performance with other IRL methods than GAIL at training time.

2. For the zero-shot transition setting in section 4.3, IMPLANT is only robust to the noise with small sigma. It would be better to explain why the IMPLANT can achieve this level of robustness and which issue limits the performance in the small perturbation. This might help the authors to further improve the robustness of IMPLANT.

3. Although the proposed method has been evaluated in diverse settings, the environments are limited to three continuous control tasks. It would be more convincing if the authors can provide more cases with both continuous and discrete action space in the rebuttal period.

#######################################################################

Questions during rebuttal period:

Please address and clarify the cons above.

---

> ### Author Response · Authors · 2020-11-21
> **Response to reviewer questions and feedback**
>
> We thank the reviewer for the constructive feedback, and we’ve conducted additional experiments to address them.
>
> * Re: Motivation for using planning
> As introduced in Sec 1, “Conceptually, IMPLANT aims to counteract the imperfections due to policy optimization in the RL step by using the reward function estimated in the IRL step for decision-time planning". Put differently, errors for imitation learning could arise either during reward inference (IRL step) or policy learning (RL step). Planning using the learned policy as an IMPLANT is a hybrid approach.
> * Re: other IRL methods
> We also experiment with PWIL [https://arxiv.org/abs/2006.04678] as our IRL algorithm for training. The result shows that our algorithm still outperforms all of the baselines in Hopper both in non-transfer setting (sigma=0.0) and in the presence of perturbations at test-time:
> Results (motor noise):
> | sigma          	| 0.0 	| 0.1 	| 0.2 	| 0.5 	| 1.0 	|
> |----------------	|-----	|-----	|-----	|-----	|-----	|
> | PWIL           	|3552 ± 200 	|2189 ± 1236	|935 ± 706	|**393 ± 119**	|218 ± 86	|
> | IMPLANT (ours) 	|**3563 ± 87** 	|**2637 ± 1140**	|**1110 ± 817**	|390 ± 103	|**232 ± 82**	|
>
> * "Why the IMPLANT can achieve this(noise with small sigma) level of robustness and which issue limits the performance in the small perturbation?"
>  A close-loop planner helps correct small mistakes that the policy makes earlier when there is moderate noise in the test environment. At each timestep, the agent has multiple candidate actions sampled from the stochastic policy instead of one, thereby offering additional opportunity to mitigate the deviations from the optimal trajectory.
> For a very high noise level, any algorithm will naturally deteriorate in performance due to a significant shift in training and testing
> dynamics.
>
> * Re: More tasks
> We experimented with one more continuous control task: Humanoid, and two discrete tasks: MountainCar and LunarLander, in the non-transfer setting(train dynamics matches test dynamics). We ran our baselines and IMPLANT on the Humanoid dataset obtained from stable-baselines [https://github.com/openai/baselines/tree/master/baselines/gail]. The results are shown below:
> | Expert          	| 588 	|
> |----------------	|-----	|
> | BC           	| 405 ± 95 	|
> | BC-Dropout	| 373 ± 49	|
> | GAIL		| 454 ± 108	|
> | IMPLANT (ours) 	| **1041 ± 603** 	|
>
>     For our discrete tasks, GAIL was able to recover the optimal policy with one expert demonstration, leaving little room for improvement in the non-transfer setting. Moreover, our problem setting largely resembles sim2real transfer, which primarily pertains to robotics. Thus, we are mainly interested in continuous action space.
>
> We hope that the above comments have answered the reviewer's questions. We appreciate the reviewer’s feedback and welcome any further comments!

---

### Official Review · AnonReviewer3 · 2020-10-28

**Rating:** 4
**Confidence:** 5

**Review:**

The authors propose a method to enhance imitation learning by using MPC on the reward function learned by an imitation learning approach. The chosen MPC approach uses the learned policy to generate candidates for a search process that maximizes the reward. To learn the reward function, the authors propose to use GAIL. Besides showing improved performance in a typical benchmark scenario, the authors compare their method with GAIL in various scenarios where some form of transfer is required. The writing is generally clear and easy to understand.

Historically, IRL methods relied on given or learned transition models as they relied on solving for a policy in an inner loop. Recently, we have seen a handful of methods introduced which do not have that short-coming and with it, we have seen a shift towards model-free methods. It is generally understood in IRL that the learned reward function can be used with any reinforcement learning or with any planning algorithm such as MPC but while some model-based approaches have been introduced, the merits of model-based imitation learning have not been thoroughly compared to recent model-free methods to my knowledge. While this makes the paper relevant, the use of a single existing MPC approach with a learned reward function is a relatively minor contribution. My main issue, however, lies in the evaluation as the dynamics model is generally just given to the agent. While the authors acknowledge that the dynamics model could be learned, that is not the case in any of the experiments and it is of little surprise that a MPC approach with perfect knowledge of the transition dynamics is able to outperform reinforcement learning approaches that have to learn from interaction. Furthermore, the authors claim their approach to be zero-shot in the transfer-experiments in comparison to AIRL. This claim is hardly defensible when IMPLANT is handed perfect knowledge of the new dynamics while AIRL has no knowledge of the new dynamics.

Another major issue with this work stems from an apparent misunderstanding of the role of the discriminator in GAIL. While GAIL treats the discriminator as a reward function in an inner loop, it is not a real IRL method. As the policy converges to the expert’s policy, the learned discriminator can no longer tell the data apart and converges to a constant. On a practical level, the proposed MPC approach uses the learned policy to generate proposals and therefore corresponds more to an iterative step on this policy. This explains why the agent is able to achieve higher scores on benchmark tasks; however, the discriminator by itself is not generally useable as a reward function like the authors suggest. This is underlined by the results that “GAIL - reward only” fails to learn. Instead, the authors should use true IRL methods as the basis for their approach.

Finally, the proposed evaluation is limited as the authors choose 3 of the easiest tasks from the mujoco benchmark. The lack of results on humanoid as the hardest task is problematic.

---

> ### Author Response · Authors · 2020-11-21
> **Response to reviewer questions and feedback**
>
> We thank the reviewer for the feedback and would like to address the following points:
> * “It is little surprise that an MPC approach with perfect knowledge of the transition dynamics is able to outperform reinforcement learning approaches that have to learn from interactions.”
> We respectfully disagree. It is not guaranteed that an MPC approach with access to transition dynamics outperforms model-free methods. For instance, the “Gail-Reward Only” baseline, which uses a random policy, fails to outperform GAIL or BC in all environments. Moreover, GAIL training requires a large number of transitions(~10millions for each environment), which is costly itself and can be utilized by training a dynamics model. Last, we believe the assumption is reasonable given a possible application of our setting - sim2real - also assumes the knowledge of a simulator.
>
> * “IMPLANT is handed perfect knowledge of the new dynamics while AIRL has no knowledge of the new dynamics”
> We apologize for the confusion, but we’ve modified the paper to clarify that we do not require interactions with the *test* environment. Instead, we use the training dynamics to plan at test-time, and even with a mismatched dynamics model, our approach still achieves higher performance than a direct policy transfer approach. Thus, our approach is zero-shot. In contrast, AIRL reoptimizes the learned reward function at test-time to perform RL, which indeed requires interactions with the test environment.
>
> * GAIL does not learn real IRL reward
> We agree with the theoretical argument, but this is not true in practice. Empirically, neither the generator nor the discriminator is perfect, especially when trained with limited expert trajectories (Sec 4.1) or in presence of nuisance variables and noise (sec 4.2). With regards to soundness, note that IMPLANT with an optimal policy and discriminator (and hence, constant discriminator output for reward) reduces to GAIL. Since GAIL is assumed to be optimal here, so is IMPLANT by corollary.
> More importantly, IMPLANT can be employed over any existing IRL approach. To demonstrate this empirically, we tested IMPLANT on a recent state-of-the-art non-adversarial IRL algorithm,  PWIL [https://arxiv.org/abs/2006.04678]. The results below show that our algorithm still outperforms all of the baselines in Hopper both in the non-transfer setting (sigma=0.0) and in the presence of motor noise perturbation at test-time:
> Results (motor noise):
> | sigma          	| 0.0 	| 0.1 	| 0.2 	| 0.5 	| 1.0 	|
> |----------------	|-----	|-----	|-----	|-----	|-----	|
> | PWIL           	|3552 ± 200 	|2189 ± 1236	|935 ± 706	|**393 ± 119**	|218 ± 86	|
> | IMPLANT (ours) 	|**3563 ± 87**	|**2637 ± 1140**	|**1110 ± 817**	|390 ± 103	|**232 ± 82**	|
> * Humanoid experiments
> We ran our baselines and IMPLANT on the Humanoid dataset obtained from stable-baselines [https://github.com/openai/baselines/tree/master/baselines/gail]. The results are shown below:
> | Expert          	| 588 	|
> |----------------	|-----	|
> | BC           	| 405 ± 95 	|
> | BC-Dropout	| 373 ± 49	|
> | GAIL		| 454 ± 108	|
> | IMPLANT (ours) 	| **1041 ± 603** 	|
>
> We hope that the above comments can clarify our setting and show the contributions of our work. We appreciate the reviewer’s feedback and welcome any further comments!

---

### Official Review · AnonReviewer4 · 2020-10-30

**Rating:** 4
**Confidence:** 5

**Review:**

IMITATION LEARNING REVIEW
============================

The gist of the paper is to improve IRL with the so-called IMPLANT algorithm which is supposed to improve control policies in learning controllers.

PROS
====
It would be nice to have a framework for minimizing model estimation errors in imitation learning. This paper presents a framework for mitigating the learned dynamics during imitation learning using an augmented cost fuction at test time. The authors called this IMPLANT>

CONS
====

I do not see how this algorithm is an MPC framework: where is the action/control law computed in lines 7-16 of Algorithm 1?
What did the authors mean in line 11 when they wrote, "First action a_0^{(i)} ~ \pi_\theta ... ".

* How was \pi in line 11 computed?

* Is there an upper bound on the length of the horizon H?

* If there is, how does the policy perform for slowly-varying trajectories at lower horizons versus fast-changing trajectories at longer planning horizons?


* I am concerned that the reward function in equation (4) is just a restatement of the optimality principle and do not see why this is being rebranded as a new algorithm.

* Of course, the rollout policy would guarantee better performance. What is the predicate for choosing a *random rollout policy* as the authors mentioned on line 4? What is the point of using a mixture?

* Since you mentioned that the rollouts can be parallel-wise executed, did you give any thoughts to the fact that the control law has to be tightly scheduled in the feedback before the next iteration of the MPC is run?

Grammar errors
==============

There were a few sentence structures in the paper that could use some revision e.g. "A" popular class of approaches"-->approach

---

> ### Author Response · Authors · 2020-11-20
> **Response to reviewer questions and feedback**
>
> We thank the reviewer for the helpful comments.
>
> * Q: where is the action/control law computed in lines 7-16 of Algorithm 1?
> A: The best action is picked in Line 13 based on returns computed using Eq. 4.  In line 8 of Algorithm 1, the agent is given a state $s_0$ from which to plan using the approach in lines 10-15. Then, the agent executes action $a^{(i^\ast)}_0$ and observes the next state $s$ from the test environment, which closes our planning loop. Thus, the whole process is an MPC.
>
> * Q: What did the authors mean in line 11 when they wrote, "First action $a_0^{(i)} \sim \pi_\theta$ ... "?
> A: In Line 11, we rollout B trajectories of length H. For each trajectory i, the first action $a_0^{(i)}$ at state $s_0$ is sampled from the policy $ \pi_\theta$.  The subsequent $(H-1)$ actions in the trajectory for timesteps > 0, denoted as $a_{>0}^{(i)}$, are sampled from the policy $\pi$.
>
> * Q: How was $\pi$ computed?
> A: The rollout policy $\pi$ is an input to the "Plan" function. In our experiments, we found it best to specify $\pi$ to be the same as the parameterized policy $\pi_\theta$ (we also tried setting $\pi_\theta$ and $\pi$  to be a random policy). In general, the policy from which we sample the initial action ($\pi_\theta$) does not have to be the same as the rollout policy ($\pi$).
> * Q: Is there an upper bound on the length of the horizon H? Difference between large H and small H?
> A: The upper bound of horizon H would just be the upper bound of a trajectory, which is 1000 in our experimental setups. We did an ablation study on how the performance of our method varies with respect to different planning horizons, which is included in Section 4.4.
> * Q: What is the predicate for choosing a random rollout policy on line 4? What is the point of using a mixture?
> A: We assume that the reviewer is referring to the “GAIL-Reward Only” baseline. Adding a random rollout policy baseline aims to demonstrate the fact that leveraging both the policy (generator) and reward function (discriminator) is indeed necessary for the algorithm to perform well; missing either will result in degradation in performance. We include the option of using a mixture because we believe there is a tradeoff in exploration between a parametric policy and a random policy. We find that this is not the case in the environments tested but could apply to other tasks especially when the parametric policy is highly biased w.r.t. the optimal policy in the test environment.
> * Q: scheduling of the control law during parallel execution of rollouts?
> A: The parallelization of rollouts during planning is indeed bottlenecked by the rollout with the largest horizon. In all of our experiments, we perform rollouts of fixed length and the horizon that corresponds to the optimal performance is not very large (10~50). Thus, the gains due to parallelization are significant. We have included this discussion in the revised draft of the paper.
>
> * Grammar Errors:
> We thank the reviewer for pointing out those errors, and we’ve modified the pdf to fix all identifiable grammar errors.

---

### Author Response · Authors · 2020-11-25
**Summary of our rebuttals**

We thank all the reviewers for their constructive comments! We would like to offer a summary of our rebuttals to address some common questions as well as highlight our contribution.

* Re: Additional assumptions of our method
As mentioned in our individual rebuttals, our method aims to mitigate the perturbations in the test time environment by planning with and IRL rewards and the *training* dynamics in a zero-shot manner. Thus, unlike IRL+reoptimizing methods such as AIRL, we do not allow interactions in the test environment, and we do not assume access to the test time environment.

* Re: Choice of IRL algorithm for our method
As shown in Algorithm 1, we can bootstrap on any IRL methods that output a policy and a reward function, which does not limit to GAIL as the one used in our experiments. We used GAIL because it is the most popular adversarial IL algorithm in use today, and empirically showed that its proxy reward function is suitable for MPC planning. We also conducted experiments with another SOTA method: PWIL and showed that our method still improved in performance.

* Re: Contributions of our method
Last, we’d like to point out that our method is a novel approach that has demonstrated its effectiveness in coping with challenging problems such as causal confusion and noises. It is not obvious that planning on top of a learned reward function can achieve higher performance, especially in those challenging transfer settings that we experimented on.

We hope that we’ve addressed all the questions and concerns from the reviewers, and we really appreciate the reviewers for providing us with helpful feedbacks that strengthened our paper.

---

### Decision · Program_Chairs · 2021-01-07
**Final Decision**

**Decision:**

Reject

**Comment:**

The reviewers highly appreciated the replies and the additional experiments. We also had a private discussion on the paper. To summarize: the replies alleviated quite a few concerns, however the consensus was that the paper still does not meet the bar for a highly competitive conference like ICLR.

The idea of combining MPC (on a 'wrong' model)  with a learned cost function is very interesting and a promising direction. On the downside the reviewers are still not entirely convinced about the contribution and believe that the paper requires a significant re-write to incorporate the discussed points as well as an additional round of reviews.